# Alkaline water as a potential agent for biting midge control: Managing effectiveness and non-target organism impact evaluation

Siti Latifatus Siriyah[1,2], I-Min Tso[1] *

1 Department of Life Science, Tunghai University, Taichung, Taiwan, 2 Department of Agrotechnology, Universitas Singaperbangsa Karawang, Karawang, Indonesia

☯ These authors contributed equally to this work.
* spider@thu.edu.tw

**Data Availability Statement:** All relevant data are within the paper and its Supporting Information files.

**Funding:** National Science and Technology Council, Taiwan Grant (NSTC 112-2327-B-029-

## Abstract

Biting midge *Forcipomyia taiwana* is one of the common pests in East Asia. Their nuisance and blood-sucking behavior causes problems not only for human health but also for some industries. This study aims to evaluate the effectiveness of spraying alkaline water on controlling biting midge population and potential side effects of such approach on non-target organisms. Laboratory experiments were conducted to evaluate the effect of alkaline water on oviposition site preference of female biting midges as well as crickets. Effect of alkaline water on distribution pattern of earthworms was also examined. Besides, we also performed field manipulative studies by long term spraying of alkaline water to evaluate the effects on biting midge density, microalgae abundance and ground arthropod communities. The results of laboratory experiments showed that female biting midges laid significantly fewer eggs in surface treated with alkaline water. However, alkaline water treatment did not significantly affect the oviposition site choice of crickets and distribution pattern of earthworms. Result of field manipulations showed that long-term spraying of alkaline water could significantly reduce the abundance of soil microalgae and density of biting midges, but did not affect the diversity of non-target ground arthropods. These results demonstrate that long-term spraying of alkaline water could decrease biting midge density without harming co-existing non-target organisms and therefore is a potentially eco-friendly approach to control such pest.

## Introduction

Biting midge *Forcipomyia taiwana* (Shiraki) (Diptera, Ceratopogonidae) was first reported by Shiraki in 1913 in central Taiwan. This species widely distributes in East Asia such as Taiwan and southern areas of China [1, 2]. Their larva feed on blue-green algae or green algae so high population density tend to occur in soil covered with abundant microalgae-containing humus [1, 3–5]. *F. taiwana* is a hematophagous insect and the females require blood as the nutrient for their egg development. Among members of the Family Ceratopogonidae, consuming mammal blood is common for adult females, but *F. taiwana* particularly prefers human blood.

001). The Directorate General of Higher Education, Ministry of Education of the Republic of Indonesia. The funders had no role in study design, data collection and analyses, decision to publish, or preparation of the manuscript.

**Competing interests:** The authors have declared that no competing interests exist.

This species actively searches human for blood during daytime, and bites exposed human skin in body parts such as legs and arms [1, 6]. This behavior not only causes nuisance for human, the bite causes serious itching and sometimes severe allergic responses depending on skin's sensitivity [2]. Even though there is still no evidence that this species can transmit disease-related viruses, microorganisms or pathogens [7], their dense abundance in outdoor areas becomes a public health concern [4], and their outbreaks have already caused economic loss for some outdoor tourism businesses [4, 8]

In the past, *F. taiwana* was not classified as a pest. However, since their outbreak in late 1980s, this species constituted a severe pest in East Asian regions such as Taiwan and has caused serious problems in certain open green spaces such as parks and schools [2, 4]. Outbreaks of biting midges also influenced the willingness of people to visit certain recreation areas and generated considerable commercial loss [9]. The pest status of biting midge is related to its overpopulation, which might be resulting from the increase in human population and activities. The increase in human population results in habitat changes and large area of natural habitats are converted into agricultural or urban areas. Transforming natural forests into bamboo, tea, or betel nut monoculture plantations creates favorable environmental conditions for biting midges. In addition, transforming natural habitats into urban green spaces such as parks and schools also creates suitable habitats for *F. taiwana*. Recently, more and more people like to perform outdoor activities such as hiking, exercising, or sightseeing in recreation areas and such activities can also enhance biting midge abundance because the gathering of people provides abundant food sources for females [2, 4, 6, 10].

It is challenging to manage hematophagous insect pests such as Ceratopogonidae and some approaches were used to suppress their populations [11]. The most common biting midge control method is using chemical pesticides. However, extensive application of insecticidal chemicals has many side effects and is harmful to human health [12–17]. Many insect repellents may cause symptoms such as breathing problems, eye and skin irritation, cough, sneezing, headache, asthma, and itching [18]. The large-scale use of insecticides not only causes human health problems, such approach also generates harmful effects to non-target organisms in the ecosystem. For example, picaridin is the main component of many insect repellents and such chemical can cause mortality of amphibians [13]. Since amphibians are one of the crucial predators of mosquito larvae, picaridin-based insect repellents actually decrease abundance of the natural enemies of target pests. In addition, most synthetic pesticides are only effective for adult pests, and the larvae or pupae of target pests are not affected. Despite the considerable amount of efforts invested, in many areas biting midge population control is far from successful [10]. Therefore, finding a new strategy that is effective and environmentally friendly is urgently needed [16, 19].

In this study, we evaluated whether biting midge population could be controlled through reducing the food resource of their larvae. The larvae of biting midges feed on microalgae and availability of microalgae may affect the density of adults [5]. Therefore, preventing massive proliferation and spread of biting midges can potentially be achieved by managing the microhabitats of microalgae to reduce their abundance [4]. The growth of microalgae in the soil is influenced by several factors such as pH. The result of a previous study showed that the growth of microalgae *Chlorella sorokiniana* declined when it was kept in the alkaline condition [20]. Since pH is one of the vital environmental factors affecting microalgal growth, we hypothesized that appropriately increasing the pH of top soil by spraying alkaline water might be able to reduce the growth of soil microalgae and consequently result in decline of adult biting midge population. In addition to testing the aforementioned hypothesis, we also examined whether such treatment would influence other organisms in the environment. In this paper, we report the results of laboratory and field manipulative studies investigating the effectiveness

of alkaline water in controlling *F. taiwana* populations and potential impacts of such approach on non-target organisms. We found that spraying alkaline water could reduce soil microalgae abundance and adult *F. taiwana* density without generating harmful effects to non-target organisms such as ground arthropods and earthworms.

## Material and methods

### 1. Production of alkaline water

We used an ionic water generator to obtain alkaline water through electrolysis (JP-ALAK50 Ionic Water Generator, Aijiahou Co. Ltd, Nantou, Taiwan). The flow rate was adjusted to 0.5 liters per minute, and the alkaline water was stored in 20-liter plastic containers. Before use the pH value of alkaline water in each container was measured using a pH indicator (First Chemical Materials Co. Ltd, Taipei, Taiwan) to ensure that the pH value was at least 10.00. Alkaline water generated from the machine was used within 24 hours to ensure that the water meet the desired quality.

### 2. Laboratory experiments

**a. Effect of alkaline water on oviposition site choice of female biting midges.** In this part of study we evaluated whether the oviposition site choice of female biting midges would be affected by the pH of the substrate surfaces. Female biting midges were collected in the campus of Tunghai University following the methods of Chuang et. al. and Shih et. al. [2, 6]. Ten females were introduced in each of the 15 transparent boxes (L = 15cm; W = 11cm; H = 7cm). The bottom of the box was divided into two parts, each containing tissue paper moistened with either alkaline or regular water every other day as egg-laying pads. After five days, the number of eggs on different moistened tissue paper was counted.

**b. Effect of alkaline water on oviposition site choice of female crickets.** This experiment aimed to study whether pH of substrate surface influenced the oviposition site choice of ground arthropods. Crickets were chosen as the model organisms, and we followed the method of de Farias-Martins et al. [21] to set up the oviposition sites. A single gravid female cricket was placed into each of 20 rectangular transparent boxes (L = 28 cm; H = 21 cm; W = 15 cm). Before we introduced female crickets to boxes, we first kept the females and the males in the same cage to let them mate. We selected large and healthy female crickets and the length of the femur and leg was measured to represent the body size. The average femur and leg length of the crickets used was 0.937 cm and 2.363 cm respectively. Inside the box, we put two small cups (8 cm in diameter and 3 cm in depth) containing soil. One cup was sprayed with alkaline and the other regular water every other day for 60 days. The number of eggs laid by female cricket in each cup was counted every week and the total number of eggs was recorded for subsequent analyses.

**c. Effect of alkaline water on distribution pattern of earthworms.** The set up of this experiment was modified from the method designed by Mvumi et al. [22]. Each plastic tray (L = 40cm, W = 30cm, H = 10cm) was filled with soil collected from the campus of Tunghai University and was divided into four quadrants. Ten individuals of earthworms were introduced in the center of the tray, then the soil surface was covered with dry grass as mulch. The soil was moistened twice a week, two quadrants was moistened with alkaline water while the other two was moistened with regular water. After eight weeks we retrieved the earthworms from the soil and compared the number of individuals in each quadrant of the tray. The result of this experiment could help us assess whether alkaline water application would affect the distribution pattern of soil animals and consequently influence ecosystem functioning.

## 3. Field experiments

**a. Study site and application of alkaline water.** The study site was located in the campus of Tunghai University, Taichung City, Taiwan. The land surface types of the Tunghai University campus were quite diverse. Such heterogeneity might potentially generate variation in biting midge density and consequently influence the effectiveness of treatment. Therefore, we used a pairwise design while setting up the sampling plots. A total of 15 pairs of sampling plots (each 10m x 10m) were established and each pair of sampling plot was located in a particular land use surface type with historical record of high biting midge density. The distance between neighboring sampling plots of the same pair was at least 20 m to minimize the potential confounding influence of biting midges moving between neighboring plots receiving different treatment. During the field experiments, alkaline water stored in 20-liter containers was transported to the study plots. Rechargeable electric sprayers were used to apply water to all surfaces of the 10m x 10m study plots. The water treatment was conducted in two stages. In the first stage (from August to September, 2022) half of the plots received 20 liters of alkaline water (pH around 10, experimental group) while the other half received same amount of regular water (pH around 7, control group) every other day for 30 days. In the second stage (from September to October, 2022), treatments were provided every day for 30 days.

**b. Effect of alkaline water on biting midge density in the field.** The density of biting midge in each plot was estimated using human leg bait method modified from Chuang et al. and Shih et al. [2, 6]. During the density estimation, a person sat in the middle of each study plot with one calf exposed for 15 minutes. The number of biting midges attracted to the calf was counted, and the result was used to quantify the abundance in the study plot. The biting midge survey was conducted during daytime from 11:00 to 16:00 because this insect tended to reach a population peak during such time period [6]. The monitoring of biting midge was conducted three times. The first was before the field manipulation (August, 2022), the second was conducted 30 days (September, 2022) and the third was conducted 60 days (October, 2022) after the initiation of manipulation.

**c. Effect of alkaline water on soil microalgae abundance in the field.** We estimated the abundance of soil microalgae by measuring the amount of chlorophyll a in the soil surface using the method modified from Tsujimura et al. [23] and the Environmental Protection Agency, Taiwan [24]. We collected soil samples from either the control or experimental plots at the end of water spraying treatments (October, 2022). Each 10m x 10m plot was divided into 4 quadrants and in each one we collected 6 grams of surface soil from the center. The soil samples collected were mixed with 10 ml of 90% alcohol, homogenized using a vortex, boiled in 60°C water bath under dark condition for 30 minutes and centrifuged at 3200 rpm for 10 minutes. Then 3 ml of supernatant was taken and the absorption readings at wavelengths 665 nm and 750 nm were recorded (designated as $E665a$ and $E750a$) using a spectrophotometer (Unico Spectrophotometer Model: UV2150, United Product & Instruments Inc.). After the initial measurements the solution was added with 0.03 ml of HCL and absorption readings at 665 nm and 750 nm were taken again (designated as $E665b$ and $E750b$). We then calculated corrected absorption readings $C665a$ and $C665b$ by the following equations [24, 25]:

$$C665a = E665a - E750a$$

$$C665b = E665b - E750b$$

The concentration of chlorophyll a ($Ca$) in soil sample was calculated by the following equation:

$$Ca = 29.62(C665a - C665b) \; x \; Ve/Vs$$

Where $Ve$ was the volume of alcohol and $Vs$ was the weight of soil sample.

**d. Effect of alkaline water on ground arthropod diversity in the field.** The diversity of ground arthropods in the study plots was estimated using ramp pitfall traps [26]. Such a trap was composed of a plastic container with a lid on top and two ramps on two sides made of transparent plastic sheets. Each container was filled with 70% alcohol up to 1.5 cm in depth, and the trap was placed in the center of the study plot for three days. The ground arthropods collected were washed and stored in glass vials containing 70% alcohol. All specimens collected were counted and identified to taxonomic order. The first ground arthropod diversity survey was conducted before the field manipulation (August, 2022), and the second was conducted 60 days after the initiation of the treatments when the manipulations were completed (October, 2022).

## 4. Statistical analyses

**a. Laboratory experiments.** Results of Shapiro wilk normality tests showed that the number of eggs laid by biting midges and crickets on surfaces moistened with water of different pH values followed normal distribution, so paired t-tests were used to compare the differences. Results of Shapiro wilk normality tests showed that earthworm abundance data also followed normal distribution, so we used a paired t-test to compare the number of earthworms in quadrants treated with either regular or alkaline water.

**b. Field experiments.** Results of Shapiro wilk normality tests showed that biting midge abundance data collected from sampling plots did not follow normal distribution, therefore we first performed log (n + 1) transformation to reduce data heterogeneity then used Wilcoxson signed rank tests to compare the densities of biting midges in study plots before and after treatments. For chlorophyll a data, we first performed a square root transformation and results of Shapiro wilk normality tests showed that the transformed data followed normal distribution. Then a paired t-test was used to compare the abundance of microalgae in plots receiving different treatments.

To compare the ground arthropod diversity between study plots receiving different treatments, we first square root transformed the abundance of all taxonomic orders of arthropods collected and Bray-Curtis similarity indices between each pair of plots were calculated. We then constructed a multidimensional scaling (MDS) plot to visualize the grouping pattern of sampling plots receiving experimental and control treatments. A permutational multivariate analysis of variance (PERMANOVA) was conducted to compare the diversity of ground arthropods collected from plots receiving different treatments while considering the effect of temporal variation. A total of 999 permutations were performed in this analysis.

All statistical analyses were performed by the R program (version 4.1.3) [27]. Shapiro-Wilk normality tests were performed by the "stats" package [28]. Box plots were made with the "ggpubr" package [29]. Wilcoxson signed rank test was conducted using the "coin" package [30]. PERMANOVA test and MDS construction were performed using the "vegan" package [31].

## Result

### 1. Laboratory experiments

**a. Effect of alkaline water on oviposition site choice of female biting midges.** In this experiment, our setup successfully made *F. taiwana* lay their eggs on the surface of moistened tissue paper. The eggs were banana-shaped and were laid singly or in groups. The eggs were initially pink and transparent and became black or brownish due to melanization [10, 32]. More eggs were laid on tissue paper surface moistened with regular water. The result of paired t-test showed that number of eggs laid on two types of surfaces was significantly different ($t = 2.03$, $p = 0.03$). The number of eggs laid on surfaces treated with regular water was almost twice that laid on alkaline surface (Fig 1).

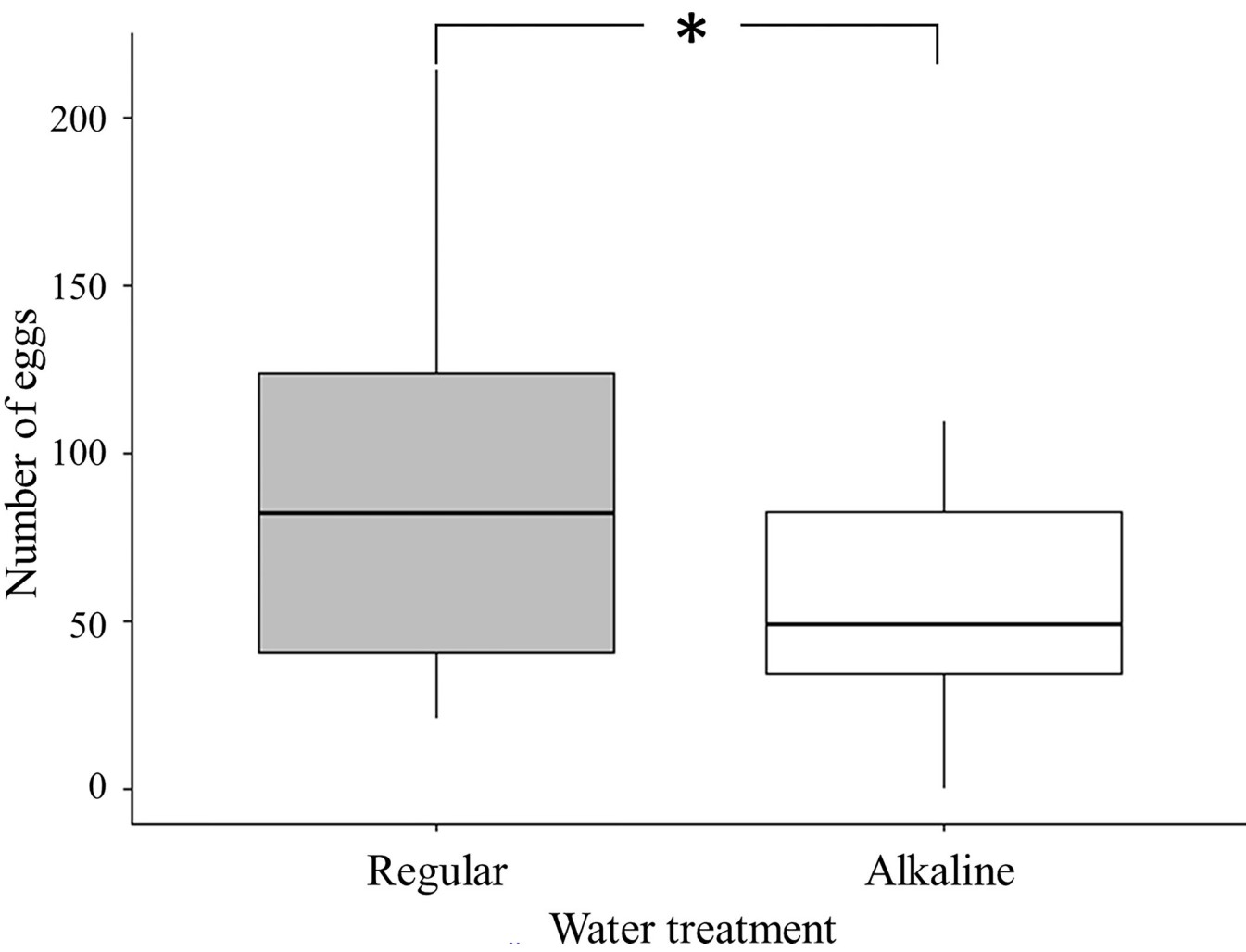

**Fig 1. Box plots of number of eggs laid by female *F. taiwana* on substrates moistened with either regular or alkaline water.** (*: $p < 0.05$).

**b. Effect of alkaline water on oviposition site choice of female crickets.** Female crickets individually kept in boxes randomly laid eggs on soil surfaces receiving either normal or alkaline water treatment. Result of paired t-test showed that no significant difference was found in number of eggs laid on two types of cups ($t = 91$, $p = 0.89$) (Fig 2).

**c. Effect of alkaline water on distribution pattern of earthworms.** At the beginning of the experiment, 10 earthworms were introduced to each tray. After 8 weeks, the number of earthworms increased several times, indicating that the microenvironments of the experimental set up were suitable for proliferation of earthworms. Results of paired t-test showed that the number of earthworms in quadrants receiving different water treatments did not differ significantly ($t = 229$, $p = 0.95$) (Fig 3). Such result indicated that alkaline water treatment did not influence earthworm proliferation and distribution patterns.

## 2. Field experiments

**a. Effect of alkaline water on biting midge density in the field.** Before the field manipulations the density of biting midges in plots designated to receive either control or experimental treatments did not differ significantly (Wilcoxson signed rank test, $V = 51.5$, $p = 0.35$)

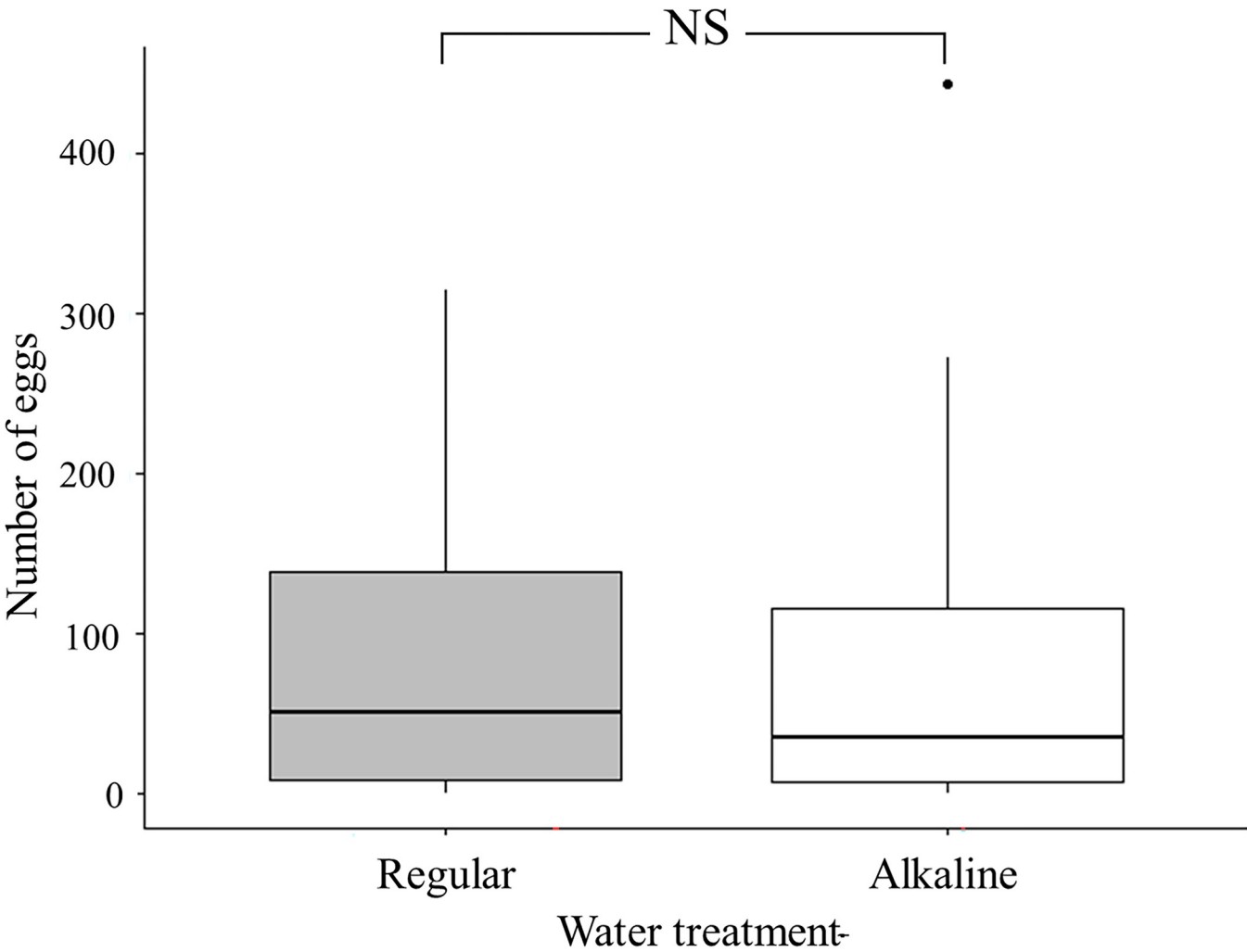

**Fig 2. Box plots of number of eggs laid by female crickets on soil surfaces moisturized with either regular or alkaline water.** (NS: non-significant).

(Fig 4). After one month of spraying water every other day, the density of biting midges in plots receiving alkaline water was significantly lower than that in plots receiving normal water (Wilcoxson signed rank test, $V = 64.5$, $p = 0.04$) (Fig 4). After another month of spraying water on a daily basis, the overall density of biting midge in all plots declined. However, the biting midge density in plots receiving alkaline water treatment was significantly lower than that of plots receiving regular water and the difference increased further (Wilcoxson signed rank test, $V = 28$, $p = 0.02$) (Fig 4). Such result indicated that long term spraying of alkaline water could decrease biting midge density in the field.

**b. Effect of alkaline water on soil microalgae abundance in the field.** Result of paired t-test showed that the amount of chlorophyll a in plots receiving alkaline water treatment was significantly lower than that of plots treated with regular water ($t = 2.21$, $p = 0.04$) (Fig 5). Such result indicated that long term spraying of alkaline water could reduce the abundance of microalgae in soil.

**c. Effect of alkaline water on ground arthropod diversity in the field.** Results of PER-MANOVA test showed that as time went by, the diversity of ground arthropods showed a significant seasonal variation (Table 1). However, study plots receiving either normal or alkaline

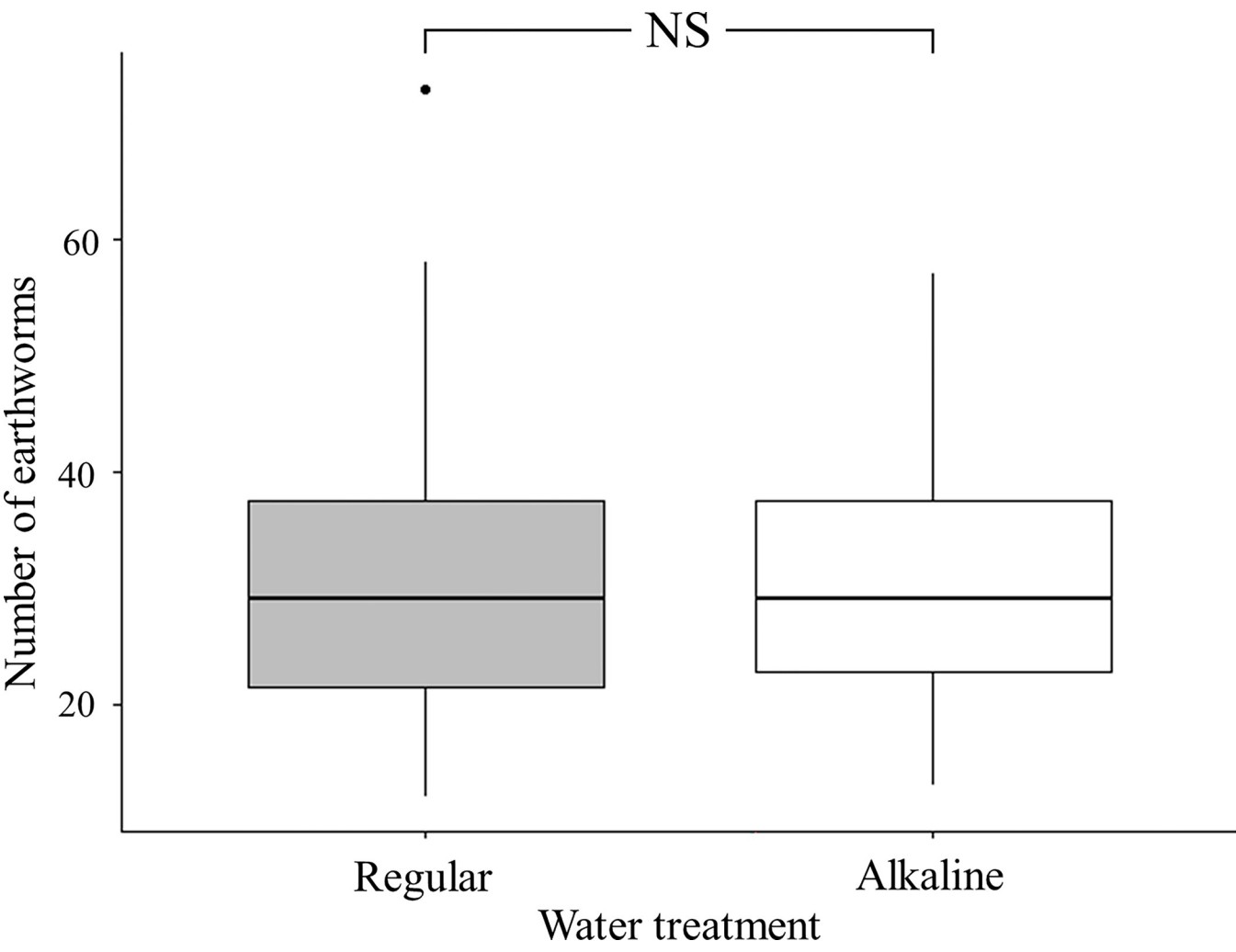

**Fig 3. Box plots of number of earthworms in tray quadrants moistened with either regular or alkaline water.** (NS: non-significant).

water treatment did not differ significantly in ground arthropod diversity (Fig 6, Table 1), even though both exhibited significant seasonal variations. Such result indicated that spraying alkaline water in the field could reduce the density of biting midges by lowering the abundance of soil microalgae, but would not influence the diversity of non-target ground arthropods.

## Discussion

In this study, we evaluated the effectiveness of alkaline water on controlling biting midge density and such method's potential impacts on non-target organisms. Results of laboratory experiments showed that number of eggs laid by female biting midges on surfaces treated with alkaline water was significantly lower, while such treatment did not affect cricket oviposition site preference or earthworm movement pattern. Results of field experiment showed that plots receiving alkaline water treatment had lower soil microalgae and biting midge density. However, ground arthropod diversity did not differ between plots treated with alkaline or regular water. Therefore, alkaline water seems to be able to reduce biting midge density without causing negative impacts on ground or soil animals.

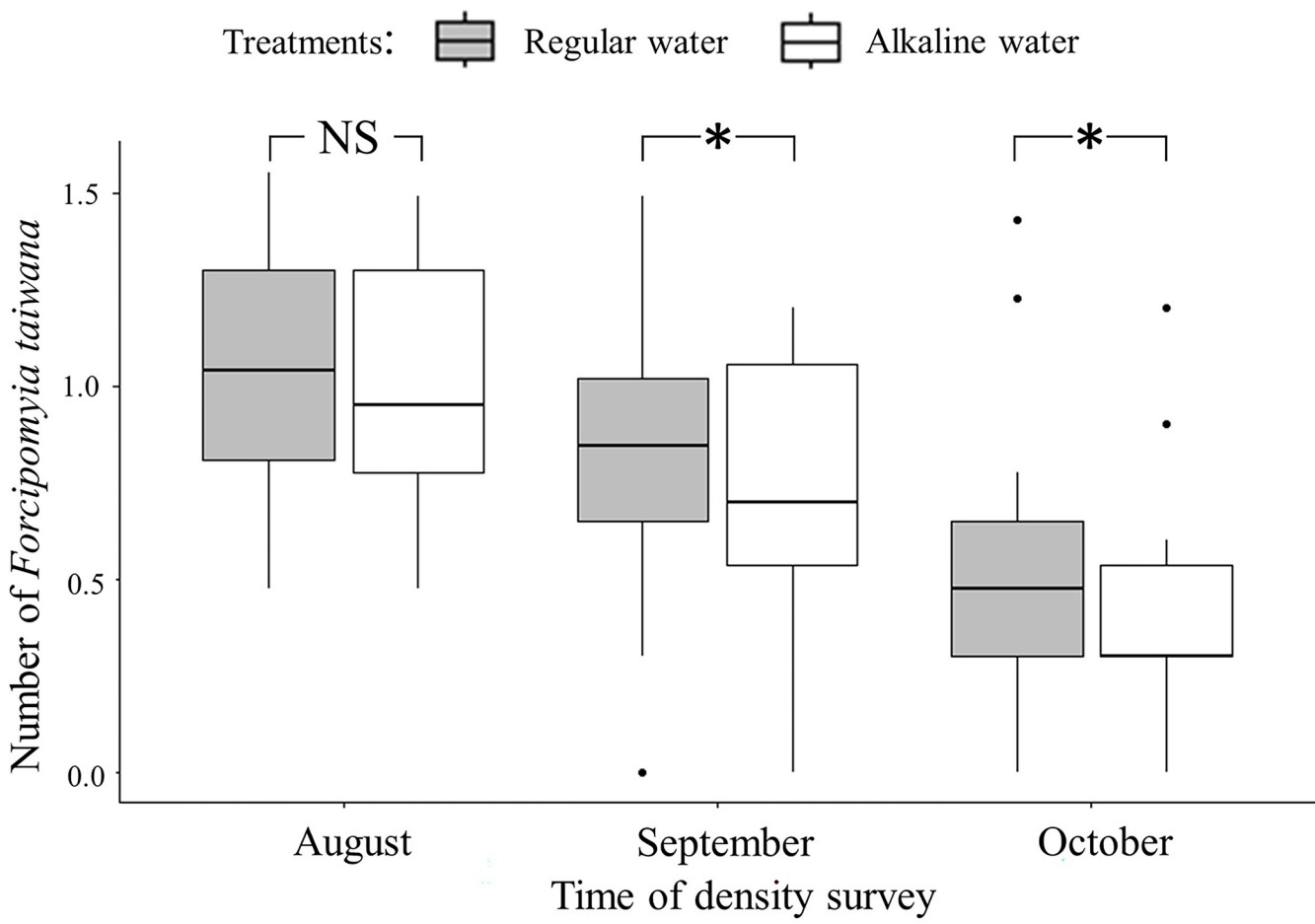

**Fig 4. Box plots of biting midge density (number per 100 m²) in study plots receiving different treatments recorded from different time periods.**

Results of our laboratory experiments showed that when female biting midges were given substrates moistened with either alkaline or regular water, they laid much fewer eggs on the former than on the latter. Such results indicated that female *F. taiwana* seemed to exhibit innate preference for oviposition sites with lower pH value. Results of previous studies showed that under alkaline condition the growth of microalgae would be negatively affected [20]. Results of our field experiments also showed that the abundance of microalgae in plots treated with alkaline water was significantly lower than that in plots treated with regular water. Therefore, the oviposition site choice pattern observed in this present study indicated that female *F. taiwana* would lay eggs in sites potentially exhibiting more food resource for their offspring.

Results of our field experiments showed that long term application of alkaline water seemed to be an effective way of reducing biting midge density. Before the field manipulation, the biting midge densities of plots designated to receive regular or alkaline water treatment were similar. In the first stage of field experiment the study plots were sprayed with either regular or alkaline water every other day for 30 days. At the end of this operation the biting midge density of plots treated with alkaline water was significantly lower than that of plots treated with regular water. In the second stage of field experiment, we increased the intensity of treatment to spray once per day. After one month of operation the density of biting midges in plots receiving alkaline water was further reduced. During two months' field experiments biting midge

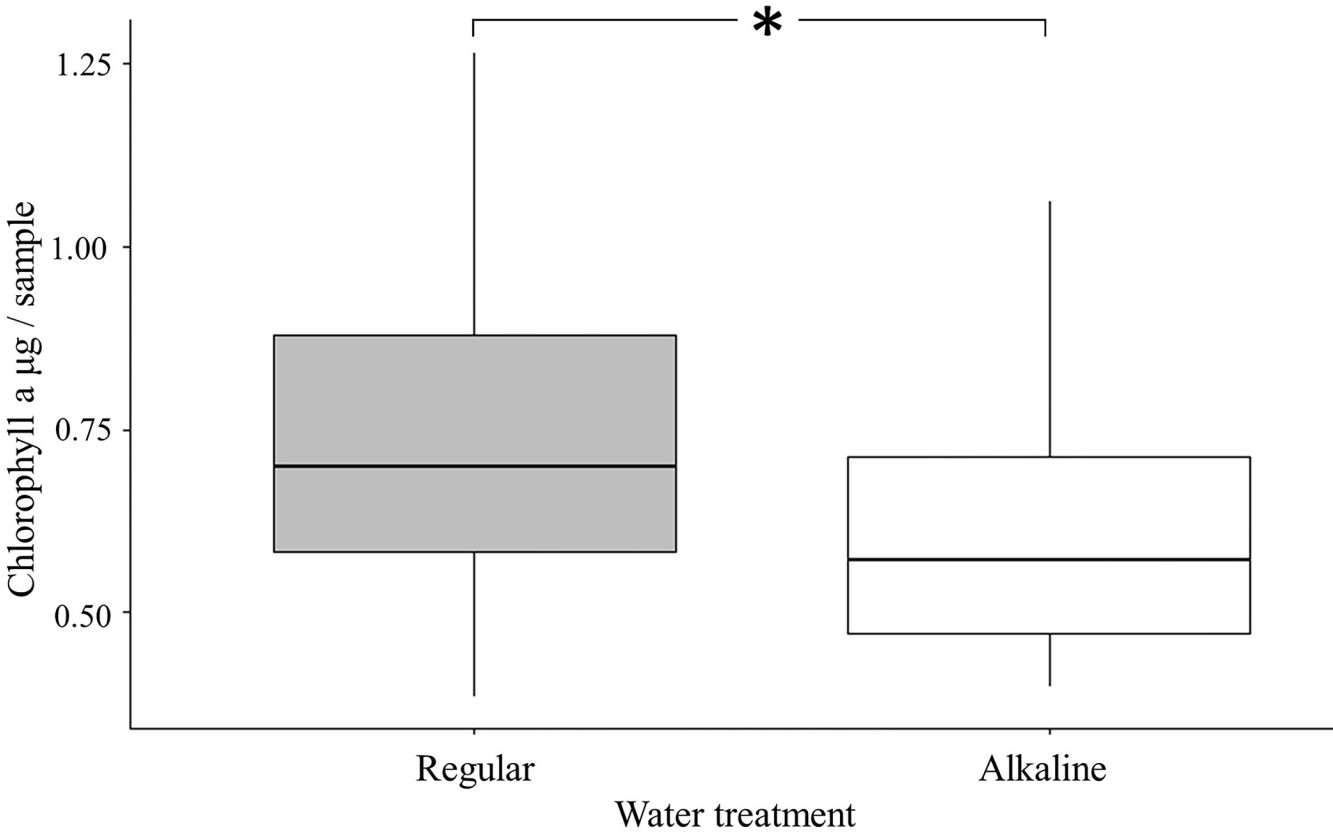

**Fig 5. Box plots of the abundance of soil microalgae estimated by concentration of chlorophyll a in study plots receiving different treatments.**

density in both types of plots exhibited a temporal decline due to seasonal factors. However, despite the influences of seasons our alkaline water treatment effectively reduced the density of *F. taiwana* and such result might be achieved by two mechanisms. First, female *F. taiwana* might lay fewer eggs in plots receiving long term spraying of alkaline water due to the higher surface soil pH value. In addition, the alkaline condition decreased the abundance of microalgae in surface soil and reduced food resource available to larvae. Therefore, our findings indicate that long term spraying of alkaline water may reduce biting midge density through influencing female oviposition site decision and reducing larval microalgae food resource.

In the past, various pesticides had been used to control the population of biting midges. Despite the effectiveness of pesticides in suppressing *F. taiwana* populations, many non-target arthropods were negatively impacted and rendered relevant pesticide approaches ecologically unfriendly. Our laboratory experiments showed that the number of eggs laid by female crickets in soil moistened with either alkaline or regular water was similar, indicating that long term

**Table 1. Results of PERMANOVA test comparing ground arthropod diversity of study plots receiving different water treatments.**

| Source of variation | df | Sum of Square | Pseudo-*F* | *P* (perm) |
|---|---|---|---|---|
| Water treatment | 1 | 0.001 | 0.97 | 0.417 |
| Season | 1 | 0.02 | 14.94 | 0.001 |
| Residual | 57 | 0.09 | | |

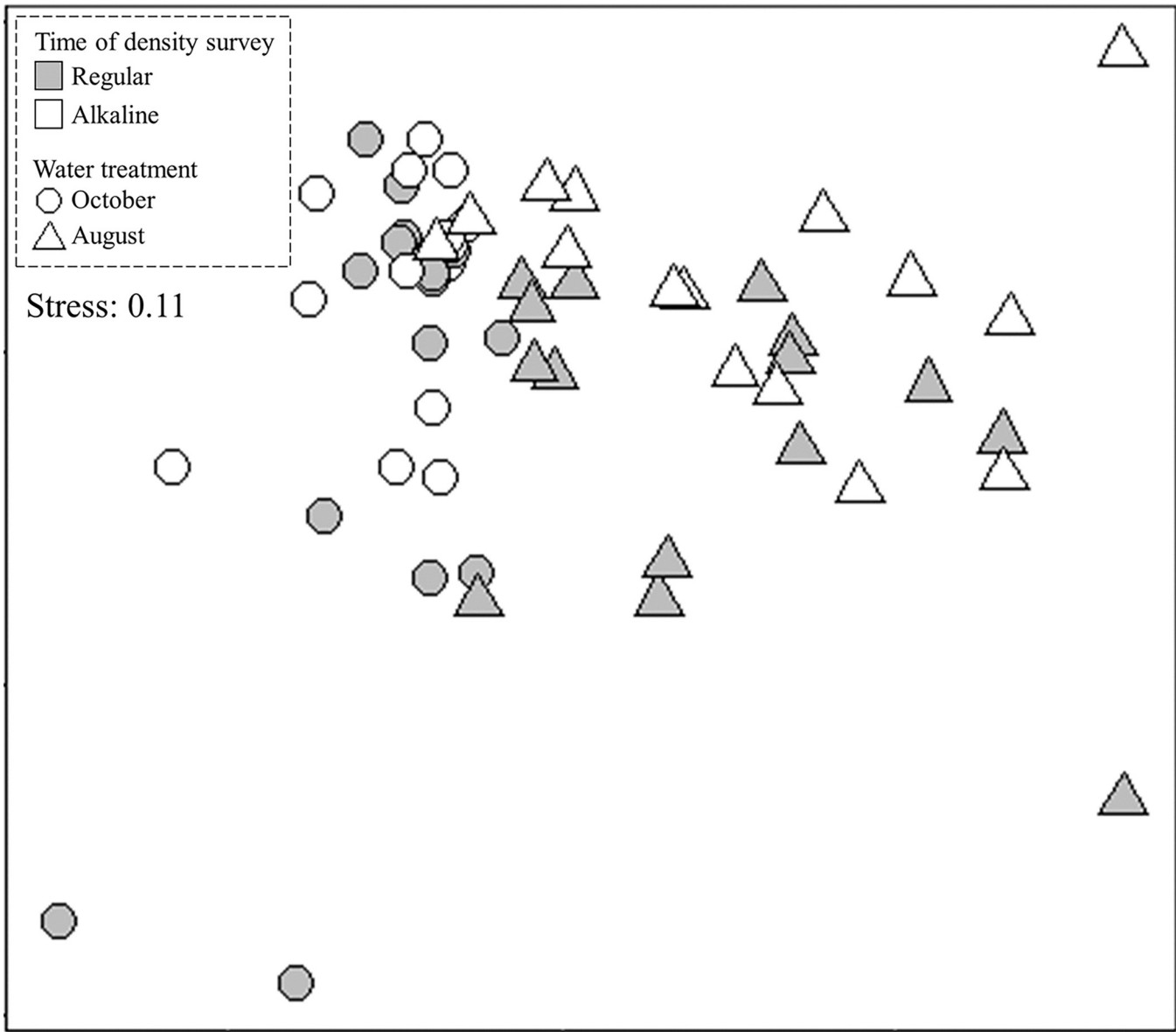

**Fig 6. Multidimensional scaling (MDS) plots of ground arthropod diversity in study plots receiving different treatments surveyed before and after field experiments.**

spraying of alkaline water to control biting midge population would not affect egg laying site selection of ground arthropods such as crickets. Congruent with the finding of this present study, results of previous studies showed that compared to pH value, soil moisture was a more critical factor in affecting oviposition site choice of crickets [21, 33]. Female crickets would lay small number of eggs when soil moisture was low and would lay no eggs when the soil moisture was too high [33]. In this present study, we did not examine the hatching rates of eggs deposited in soil receiving different treatments. Although the effects of various pH on development and hatchability of cricket eggs are poorly understood, because of the following reasons we suggest that the egg hatchability of crickets should not be affected by the alkaline water used in this study. Firstly, crickets usually deposit the eggs deep in the soil rather than on the soil surface [34], so alkaline water sprayed on the soil surface should not affect the eggs too

much. Secondly, the study conducted in New Zealand showed that population density of *Hemiandrus* sp. (weta or giant flightless cricket) was not affected by soil properties including pH [35], indicating that cricket egg hatchability and development were not sensitive to pH. We suggest that in the future the effects of alkaline water on egg hatchability and development of ground arthropods such as crickets be investigated to further confirm the feasibility of such approach.

In this present study we evaluated the effect of spraying alkaline water on non-target soil organisms such as earthworms. Earthworms are soil organisms and play important roles in soil ecosystems. Due to their thin skin, earthworms are sensitive to soil environmental pressures such as drought and metal/chemical pollutants [36, 37]. Microenvironmental factors such as temperature, pH, porosity, soil types and moisture will affect distribution and development of earthworms [37, 38]. Therefore, any eco-friendly pest control operation must ensure that activity and proliferation of earthworms are unaffected. In our laboratory experiment, earthworms were free to move between soil quadrants moistened with either regular or alkaline water. After eight weeks, the number of earthworms in soil quadrants treated with either regular or alkaline water did not differ, moreover, the abundance increased several folds. Such results indicate that long term spraying of alkaline water on soil surface to control biting midge population would not affect distribution and fecundity of soil organisms such as earthworms. One reason for such result might be that the spraying of alkaline water only altered the pH of thin top soil layer in which biting midge larva lived and fed on microalgae. The pH of deeper soil layer in which earthworms inhabited was not affected and therefore they were not influenced. Moreover, the excretion of ammonia on body surface and calciferous gland secretion in gut might potentially help neutralize the alkaline soil they contacted or ingested [39].

In this study, we also evaluated the potential impacts of applying alkaline water on ground arthropod communities. Members of ground arthropod communities such as insects and spiders serve as decomposers, predators, pollinators, soil engineers. . .etc. and play important roles in the functioning of ecosystems [40]. An eco-friendly biting midge managing approach must ensure that relevant treatments would not influence abundance and diversity of ground arthropods. Previous studies showed that many traditional pest control approaches using pesticide-based applications generated various negative impacts on arthropod communities, such as decline of beneficial arthropods like pollinators and natural enemies [12]. In this present study, before the field manipulative treatments the diversity of ground arthropods of plots designated to receive either regular or alkaline water application was similar. After two months of field experiment, ground arthropod diversities of plots receiving different treatments did not differ significantly, despite existence of seasonal arthropod community alterations. Therefore, long term application of alkaline water to manage biting midge populations seems to have little impact on ground arthropod diversity and consequently such approach can potentially ensure a normal ecosystem functioning.

Results of our study showed that a daily application of alkaline water was effective in reducing microalgae abundance and biting midge density and we consider that such protocol can be realistically applied in urban areas. Urban localities suffering from high biting midge densities are usually those exhibiting high human population density and green spaces situated between buildings and pavements such as elementary/middle school campuses and small to intermediate-sized urban parks. It is relatively easy to generate enough amount of alkaline water by using industrial-scale ionic water generators to perform daily spraying in such localities. As was fully demonstrated in our study, with appropriate logistic supports such as water tanks, transportation vehicles and powered sprayers it is possible to use alkaline water to control biting midges in outdoor areas. However, despite the effectiveness of such approach demonstrated in this present study, further studies are recommended to assess the potential

influences of various environmental factors such as habitat types, elevation, temperature, soil composition and precipitation to further confirm the feasibility of this strategy.

## Supporting information

**S1 File.**
(DOCX)

## Acknowledgments

We would like to thank Wan-Yu Wu for helping with ground arthropod identification and statistical analyses. Wen-Shen Wu and Yun-Xiao Hsiah helped with the logistics and spraying of water in the field. Shao-Lun Liu and Wei Bai helped with soil microalgae abundance analyses. We also would like to thank Yang-Sin Lo of Center for Environmental Protection and Occupational Safety and Health, Tunghai University for various logistic assistances.

## Author Contributions

**Conceptualization:** I-Min Tso.

**Data curation:** Siti Latifatus Siriyah, I-Min Tso.

**Formal analysis:** Siti Latifatus Siriyah, I-Min Tso.

**Funding acquisition:** I-Min Tso.

**Investigation:** Siti Latifatus Siriyah, I-Min Tso.

**Methodology:** Siti Latifatus Siriyah, I-Min Tso.

**Project administration:** I-Min Tso.

**Resources:** I-Min Tso.

**Supervision:** I-Min Tso.

**Validation:** I-Min Tso.

**Writing – original draft:** Siti Latifatus Siriyah, I-Min Tso.

**Writing – review & editing:** Siti Latifatus Siriyah, I-Min Tso.

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
