## [Decision Letter · Decision Letter 0]

17 May 2023

PONE-D-23-06123Alkaline water as a potential agent for biting midge control: Managing effectiveness and non-target organism impact evaluationPLOS ONE

Dear Dr. Tso,

Thank you for submitting your manuscript to PLOS ONE. After careful consideration, we feel that it has merit but does not fully meet PLOS ONE’s publication criteria as it currently stands. Therefore, we invite you to submit a revised version of the manuscript that addresses the points raised during the review process.

We look forward to receiving your revised manuscript.

Kind regards,

Samuel Adelani Babarinde, PhD

Academic Editor

PLOS ONE

Journal Requirements:

“1. National Science and Technology Council, Taiwan grant (MOST 109-2311-B-029-001-MY3)

2. the Directorate General of Higher Education, Ministry of Education and Culture of The Republic of Indonesia”

“This study is partially supported by a National Science and Technology Council, Taiwan grant (MOST 109-2311-B-029-001-MY3) to I-Min Tso. We would like to thank the Directorate General of Higher Education, Ministry of Education and Culture of The Republic of Indonesia for supporting Siti Latifatus Siriyah.”

“1. National Science and Technology Council, Taiwan grant (MOST 109-2311-B-029-001-MY3)

2. the Directorate General of Higher Education, Ministry of Education and Culture of The Republic of Indonesia”

Reviewers' comments:

Reviewer's Responses to Questions

**Comments to the Author**

1. Is the manuscript technically sound, and do the data support the conclusions?

Reviewer #1: Partly

Reviewer #2: No

2. Has the statistical analysis been performed appropriately and rigorously? 

Reviewer #1: I Don't Know

Reviewer #2: No

3. Have the authors made all data underlying the findings in their manuscript fully available?

Reviewer #1: Yes

Reviewer #2: No

4. Is the manuscript presented in an intelligible fashion and written in standard English?

Reviewer #1: Yes

Reviewer #2: Yes

5. Review Comments to the Author

Reviewer #1: Thanks to the authors of “Alkaline water as a potential agent for biting midge control: Managing effectiveness and non-target organism impact evaluation”, who tried to evaluate the effectiveness of spraying alkaline water on controlling biting midges population. The results could be useful for researchers in the field of medical or veterinary entomologists.

The following suggestions could be listed.

- As this valuable study has been conducted in a local set, the authors should be more careful to avoid generalizing the results. We know that the further studies in different sets and conditions (influenced by climatically and soil characteristics), will guide us for further conclusions. The current study could be as one of the first valuable steps which should be followed by more detailed and different studies.

- Notable parts of introduction could and should be easily omitted (e.g., L6468, 86-100). These and other parts could be summarized and only the parts which are needed and are in line with the results, should be emphasized. The current version of introduction is educational and not for a scientific publication.

- Also discussion could be revised. Some parts could be removed or moved to introduction (e.g., L340-349). Some parts of discussion are the repetition of the goals of study (e.g., L380-386).

Reviewer #2: It is a good effort made by the researchers, but it cannot be applied in the field

It is impossible to use its alkaline water in the field, so it will not practically solve a problem in public health.

Please, attend to the issues raised by both reviewers. 

You may wish to highlight the scientific usefulness of the research despite the reservation raised by the second reviewer that the Results may not be applicable to field condition.

---

## [Author Response · Author response to Decision Letter 0]

31 May 2023

Responses to Journal Requirements:

Reply: In this draft we have carefully checked formatting, reference citations, reference list and spellings according to the latest PLOS ONE house style.

Reply: In our study the field experiment was conducted in the campus of our institute, Tunghai University, under the logistic support of Center for Environmental Protection and Occupational Safety and Health, Tunghai University and no permit was needed. In addition, in Taiwan no special permit is needed to study biting midges and commonly-seen invertebrates in the field.

3. 

a. Please clarify the sources of funding (financial or material support) for your study. List the grants or organizations that supported your study, including funding received from your institution.

Reply: the financial supports received by this study were as follows

National Science and Technology Council, Taiwan Grant (MOST 109-2311-B-029-001- MY3)

The Directorate General of Higher Education, Ministry of Education of the Republic of Indonesia

Reply: The funders had no role in study design, data collection and analyses, decision to publish, or preparation of the manuscript.

Reply: The author Siti Latifatus Siriyah received a scholarship (partly) from the National Science and Technology Council, Taiwan Grant (MOST 109-2311-B-029-001- MY3), and also received a scholarship from The Directorate General of Higher Education, Ministry of Education of the Republic of Indonesia

4. Please note that funding information should not appear in the Acknowledgments section or other areas of your manuscript. We will only publish funding information present in the Funding Statement section of the online submission form. Please remove any funding-related text from the manuscript and let us know how you would like to update your Funding Statement.

Reply: We had removed the funding-related text from the Acknowledgement section of this draft of manuscript. (Lines 421-426).

5. In your Data Availability statement, you have not specified where the minimal data set underlying the results described in your manuscript can be found….. All PLOS journals require that the minimal data set be made fully available…. Upon re-submitting your revised manuscript, please upload your study’s minimal underlying data set as either Supporting Information files or to a stable, public repository and include the relevant URLs, DOIs, or accession numbers within your revised cover letter.

Reply: We had provided the minimal underlying data set of our study in a Supporting Information file. 

Responses to reviewers' comments:

Reviewer 1:

1. As this valuable study has been conducted in a local set, the authors should be more careful to avoid generalizing the results. We know that the further studies in different sets and conditions (influenced by climatically and soil characteristics), will guide us for further conclusions. The current study could be as one of the first valuable steps which should be followed by more detailed and different studies.

Reply: In this study, we showed for the first time that spraying alkaline water was a feasible way to control biting midge in the field, at least in school campuses. We agree with the reviewer that our study was conducted in a local scale and it is not appropriate to use the study result generated from a single site to claim that such approach is universally effective. Therefore, in this draft we revised our conclusion according to the opinion of the reviewer by stating that despite the effectiveness of such approach in this present study, further studies are recommended to assess the potential influences of various environmental factors such as habitat types, elevation, temperature, soil composition, precipitation …etc. to further confirm the feasibility of this strategy in different areas. (Lines 414-418).

2. Notable parts of introduction could and should be easily omitted (e.g., L64-68, 86-100). These and other parts could be summarized and only the parts which are needed and are in line with the results, should be emphasized.

Reply: We followed the suggestion of the reviewer and relevant texts in the introduction section were deleted or condensed. (Lines 54-56)

3. Also discussion could be revised. Some parts could be removed or moved to introduction (e.g., L340-349). Some parts of discussion are the repetition of the goals of study (e.g., L380-386).

Reply: We followed the suggestion of the reviewer and relevant sentences were deleted from the introduction section.

Reviewer 2:

1. It is a good effort made by the researchers, but it cannot be applied in the field. It is impossible to use its alkaline water in the field, so it will not practically solve a problem in public health.

Reply: In this study, we provided empirical evidence from both laboratory and field experiments that spraying alkaline water could effectively reduce the density of biting midges without affecting non-target invertebrates important for ecosystem functioning. We consider spraying alkaline water a very feasible and applicable method form controlling biting midges in the field for the following reasons:

a. Our study sites were located in various parts of the campus of Tunghai University, Taichung City, Taiwan. The overall area of Tunghai University is about 140 ha and it has a diverse array of habitat types. We conducted manipulated studies in sites located in such heterogenous landscapes and found such method to be effective. Therefore, scientifically spraying alkaline water should be a feasible method in controlling biting midges. 

b. Results of our study showed that a daily application of 20 liters of alkaline water per 100 square meters was effective in reducing microalgae abundance and biting midge density. We consider that such protocol should be easily applicable to urban localities suffering high densities of biting midges. Urban localities exhibiting high biting midge densities are those with high human population density and green spaces. It is relatively easy to generate enough amount of alkaline water by using industrial-scale ionic water generators to perform the spraying in such localities. For example, by using a regular industrial-scale ionic water generator a total of 360 liters of alkaline water could be produced in a night, which could be used to spray 1800 square meters of green spaces. Therefore, as was fully demonstrated in our study, with appropriate logistic supports such as water tanks, transportation vehicles, powered sprayers…etc. it is possible to use alkaline water to control biting midges in outdoor areas. Therefore, for urban localities exhibiting patches of green spaces situated between buildings and pavements such as elementary/middle school campuses and small to intermediate sized urban parks alkaline water is a potentially effective and eco-friendly way to control biting midges. We had supplemented a paragraph in the discussion section to describe the feasibility of such strategy in controlling biting midges in urban areas. (Lines 405-418)

---

## [Decision Letter · Decision Letter 1]

21 Jun 2023

PONE-D-23-06123R1Alkaline water as a potential agent for biting midge control: Managing effectiveness and non-target organism impact evaluationPLOS ONE

Dear Dr. Tso,

Thank you for submitting your manuscript to PLOS ONE. After careful consideration, we feel that it has merit but does not fully meet PLOS ONE’s publication criteria as it currently stands. Therefore, we invite you to submit a revised version of the manuscript that addresses the points raised during the review process.

We look forward to receiving your revised manuscript.

Kind regards,

Samuel Adelani Babarinde, PhD

Academic Editor

PLOS ONE

Journal Requirements:

**Comments to the Author**

1. Have the authors adequately addressed your comments raised in a previous round of review

Reviewer #3: (No Response)

Reviewer #4: All comments have been addressed

2. Is the manuscript technically sound, and do the data support the conclusions?

Reviewer #3: Yes

Reviewer #4: Yes

3. Has the statistical analysis been performed appropriately and rigorously? 

Reviewer #3: Yes

Reviewer #4: Yes

4. Have the authors made all data underlying the findings in their manuscript fully available?

Reviewer #3: Yes

Reviewer #4: Yes

5. Is the manuscript presented in an intelligible fashion and written in standard English?

Reviewer #3: Yes

Reviewer #4: Yes

6. Review Comments to the Author

Reviewer #3: Thanks for this valuable paper but it needs minor revision such as tense for some verbs used in text and also in some paragraphs to be inserted in more suitable sites.

Reviewer #4: The manuscript is well written and revised. I might ask about the type of ions making water alkaline? As the authors mentioned that this water (pH 10) had no effects on earth worm life and female crickets to oviposit. They examined the eggs weekly but no result about hatchability. Such females might be insensitive to the pH so they continued to lay eggs. So if embryos suffered mal-formation or decreased hatchability so this water would affect consequently ecosystem.

Could the authors mention a sentence about hatchability of cricket’s eggs (%)?

---

## [Author Response · Author response to Decision Letter 1]

6 Jul 2023

Response to Reviewer 

Reviewer #4

1. The manuscript is well written and revised. I might ask about the type of ions making water alkaline? 

Reply: As we mentioned in the method, the alkaline water used in our field experiments was produced by an ionic water generator, and there is no need to supplement additional chemical compound to the water.

2. As the authors mentioned that this water (pH 10) had no effects on earth worm life and female crickets to oviposit. They examined the eggs weekly but no result about hatchability. Such females might be insensitive to the pH so they continued to lay eggs. So if embryos suffered mal-formation or decreased hatchability so this water would affect consequently ecosystem. Could the authors mention a sentence about hatchability of cricket’s eggs (%)?

Reply: The concern of the reviewer is reasonable because if the egg hatching of crickets is affected then the roles crickets played in ecosystems will be affected. However, because of the following reasons we suggest that the egg hatchability of crickets should not be affected by the alkaline water used in this study. Firstly, crickets usually deposit the eggs deep in the soil rather than on the soil surface. When laying the eggs, female crickets will dig the soil using their front legs (ca <5 mm deep), then the females would insert the ovipositor and half of the abdomen in the hole they dig to release eggs (Réale & Roff, 2002). Therefore, the alkaline water sprayed on the soil surface should not affect the eggs too much. On the other hand, the effects of various pH on development and hatchability of cricket eggs are poorly understood. Most relevant studies focused on the effects of temperature and drought on cricket egg development. However, the study conducted in New Zealand showed that population density of Hemiandrus sp. (weta or giant flightless cricket) was not affected by soil properties including pH (Nboyine et al., 2016). Such result suggested that cricket egg hatchability and development should not be affected by the alkaline water we used in this present study. In this draft, we follow the suggestion of the reviewer and supplement several sentences to mention about the potential effects of alkaline water on egg hatchability of crickets (lines 370 - 381).

Reference

Nboyine JA, Boyer S, Saville D, Smith MJ, Wratten SD. 2016. Ground wētā in vines of the Awatere Valley, Marlborough: biology, density and distribution. New Zealand Journal of Zoology. 43: 336–350.doi:10.1080/03014223.2016.1193548.

Réale, D., & Roff, D. A. (2002). Quantitative genetics of oviposition behaviour and interactions among oviposition traits in the sand cricket. Animal Behaviour, 64(3), 397–406. https://doi.org/10.1006/anbe.2002.3084

---

## [Decision Letter · Decision Letter 2]

4 Aug 2023

Alkaline water as a potential agent for biting midge control: Managing effectiveness and non-target organism impact evaluation

PONE-D-23-06123R2

Dear Dr. Tso,

We’re pleased to inform you that your manuscript has been judged scientifically suitable for publication and will be formally accepted for publication once it meets all outstanding technical requirements.

Kind regards,

Samuel Adelani Babarinde, PhD

Academic Editor

PLOS ONE

Additional Editor Comments (optional):

Reviewers' comments:

Reviewer's Responses to Questions

**Comments to the Author**

1. If the authors have adequately addressed your comments raised in a previous round of review and you feel that this manuscript is now acceptable for publication, you may indicate that here to bypass the “Comments to the Author” section, enter your conflict of interest statement in the “Confidential to Editor” section, and submit your "Accept" recommendation.

Reviewer #3: (No Response)

Reviewer #4: All comments have been addressed

2. Is the manuscript technically sound, and do the data support the conclusions?

Reviewer #3: Yes

Reviewer #4: Yes

3. Has the statistical analysis been performed appropriately and rigorously? 

Reviewer #3: Yes

Reviewer #4: Yes

4. Have the authors made all data underlying the findings in their manuscript fully available?

Reviewer #3: Yes

Reviewer #4: Yes

5. Is the manuscript presented in an intelligible fashion and written in standard English?

Reviewer #3: Yes

Reviewer #4: Yes

6. Review Comments to the Author

Reviewer #3: (No Response)

Reviewer #4: (No Response)

7. PLOS authors have the option to publish the peer review history of their article (what does this mean?). If published, this will include your full peer review and any attached files.

Reviewer #3: **Yes: **Walaa Abdel Moneim Elkholy

Reviewer #4: No

---

## [Editor Report · Acceptance letter]

10 Aug 2023

PONE-D-23-06123R2 

Alkaline water as a potential agent for biting midge control: Managing effectiveness and non-target organism impact evaluation 

Dear Dr. Tso:

I'm pleased to inform you that your manuscript has been deemed suitable for publication in PLOS ONE. Congratulations! Your manuscript is now with our production department. 

Kind regards, 

on behalf of

Dr. Samuel Adelani Babarinde 

Academic Editor

PLOS ONE